# Clinically translatable mitochondrial gene therapy in muscle using tandem mtZFN architecture

Pavel A Nash[1], Keira M Turner[1], Christopher A Powell [ID][1], Lindsey Van Haute [ID][1], Pedro Silva-Pinheiro[1], Felix Bubeck[2,3], Ellen Wiedtke[2,3], Eloïse Marques[1], Dylan G Ryan[1], Dirk Grimm [ID][2,3,4,5], Payam A Gammage [ID][1,6,7] & Michal Minczuk [ID][1,8 ✉]

## Abstract

**Mutations in the mitochondrial genome (mtDNA) often lead to clinical pathologies. Mitochondrially-targeted zinc finger nucleases (mtZFNs) have been successful in reducing the levels of mutation-bearing mtDNA both in vivo and in vitro, resulting in a shift in the genetic makeup of affected mitochondria and subsequently to phenotypic rescue. Given the uneven distribution in the mtDNA mutation load across tissues in patients, and a great diversity in pathogenic mutations, it is of interest to develop mutation-specific, selective gene therapies that could be delivered to particular tissues. This study demonstrates the effectiveness of in vivo mitochondrial gene therapy using a novel mtZFN architecture on skeletal muscle using adeno-associated viral (AAV) platforms in a murine model harboring a pathogenic mtDNA mutation. We observed effective reduction in mutation load of cardiac and skeletal muscle, which was accompanied by molecular phenotypic rescue. The gene therapy treatment was shown to be safe when markers of immunity and inflammation were assessed. These results highlight the potential of curative approaches for mitochondrial diseases, paving the way for targeted and effective treatments.**

**Keywords** Gene Therapy; Zinc Finger Nuclease (mtZFN); Adeno-Associated Viruses (AAV); mtDNA Heteroplasmy Modification; Skeletal Muscle
**Subject Categories** Genetics, Gene Therapy & Genetic Disease; Organelles

See also: AS Ghifari & M Ott

## Introduction

Mitochondrial diseases, resulting from mutations in the mitochondrial genome (mtDNA), have a minimum prevalence of 1 in 5000 adults (Gorman et al, 2015). Currently, these conditions remain largely untreatable and without a cure. Due to the polyploid nature of mtDNA, a mix of mutated and wild-type genomes, known as heteroplasmy, is often present in the disease state. To date, mammalian mtDNA has proven resistant to manipulation by CRISPR-based technologies given the inability of mitochondria to efficiently uptake gRNA (Gammage et al, 2018a). While direct manipulation of mammalian mtDNA through CRISPR-free base editing is gaining attention (Silva-Pinheiro and Minczuk, 2022), targeting heteroplasmy with site-specific nucleases has already acquired substantial evidence, both in vitro and in vivo, as a promising alternative therapeutic approach for these disorders (Jackson et al, 2020).

Recent studies have investigated the heteroplasmy-shifting activity of various mitochondrially targeted nucleases. These include zinc-finger nucleases (mtZFN), transcription activator-like effector nucleases (mitoTALEN), and meganucleases (mitoARCUS), delivered via adeno-associated virus (AAV) in a heteroplasmic m.5024C>T mouse model of mtDNA disease (Gammage et al, 2018b; Bacman et al, 2018; Zekonyte et al, 2021). In vivo efficacy of mitoARCUS was also demonstrated using a m.3243A>G xenograft mouse model using systemic AAV delivery (Shoop et al, 2023). These studies have illustrated the potential of heteroplasmy shifting agents in vivo for future treatment strategies. However, significant barriers hinder the translation of these approaches from animal models to patients. One major challenge is achieving safe and efficient dosing with feasible AAV titers (Silva-Pinheiro and Minczuk, 2022). In particular, for dimeric mitochondrial nucleases (mtZFN and mitoTALEN), due to constraints in AAV genome coding capacity, experimental strategies rely on the co-administration of AAV virions, each carrying sequences sufficient to encode a single nuclease monomer. However, direct translation of current co-infection strategies for classical bipartite mitoTALEN or mtZFN into patients would require AAV titers exceeding reasonable manufacturing capacity and potentially posing safety risks (Gammage et al, 2018b; Bacman et al, 2018). While mitoTALEN-based approaches face challenges due to their significant nucleic acid coding requirements, mtZFN-based strategies offer a theoretical advantage owing to the possibility for being optimized for delivery within a single AAV virion. This would constitute a significant

[1]MRC Mitochondrial Biology Unit, University of Cambridge, Cambridge, UK. [2]Department of Infectious Diseases/Virology, Section Viral Vector Technologies, Medical Faculty, Heidelberg University, Heidelberg, Germany. [3]BioQuant, BQ0030, Heidelberg University, Heidelberg, Germany. [4]Faculty of Engineering Sciences, Heidelberg University, Heidelberg, Germany. [5]German Center for Infection Research (DZIF) and German Center for Cardiovascular Research (DZHK), partner site Heidelberg, Heidelberg, Germany. [6]CRUK Scotland Institute, Glasgow, Glasgow, UK. [7]School of Cancer Sciences, University of Glasgow, Glasgow, UK. [8]Department of Clinical Neurosciences, University of Cambridge, Cambridge, UK. ✉E-mail: mam201@cam.ac.uk

technical advance, as the entire heteroplasmy-shifting apparatus can be delivered to every infected cell, eliminating the need for co-infection of separate virions, thereby reducing dosing requirements (Silva-Pinheiro and Minczuk, 2022).

To address this technical limitation, we present the design and testing of a tandem-linked mtZFN architecture. This novel architecture can be packaged into a single AAV virion, enabling effective heteroplasmy modification with lower doses of AAV and improved efficacy in shifting heteroplasmy. Our study demonstrates the potential of mtZFNs as a robust platform for correcting mtDNA mutations. The findings suggest that this approach could be safe and effective for treating mitochondrial pathologies and may be applicable to human clinical trials, representing a significant leap forward in the field of genomic medicine.

## Results

### Order of tandem mtZFNs affect heteroplasmy shifting in vitro

We aimed to generate a single recombinant AAV (rAAV) virion encoding a tandem mtZFN, specific to the heteroplasmic mutation site of the m.5024C>T mouse (Kauppila et al, 2016), containing MTM25 (mutant-specific) and WTM1 (companion construct) mtZFN monomer coding sequences (Fig. 1A) (Gammage et al, 2018b). The compact nature of the $Cys_2His_2$ motif theoretically enables the co-delivery of heterodimeric mtZFN constructs in a single rAAV. In the case of the MTM25 and WTM1 (Fig. 1A), the total sequence length is 2576 bp for both mtZFN monomers, leaving ~2 kb for additional sequence elements in cis necessary for mtZFN expression. However, expression of the constructs from the same open reading frame (ORF) requires the addition of a "self-cleaving" peptide sequence to separate both protein products upon translation. The 2A viral peptide family enables such functionality, harboring a conserved DxExNPGP motif that promotes ribosomal skipping upon translation (Liu et al, 2017). In this study, the 18 amino acid-long T2A sequence was chosen (Fig. 1B), which is among the most efficient in mammalian cells (Chng et al, 2015). To assess how tandem configuration of mtZFNs affects mitochondrial matrix delivery and stoichiometry, as compared to separate configurations, western blotting analysis was performed. To this end, cells were transfected with equimolar amounts of mtZFN-coding DNA delivered on [1] separate plasmids, [2] separate plasmids with attenuation of expression using the hammerhead ribozyme (HHR), which destabilizes mRNA transcripts (Gammage et al, 2016), and [3] tandem configurations using T2A in both sequential arrangements of mtZFN monomers (WTM1-T2A-MTM25 or MTM25-T2A-WTM1) (Fig. 2A). Assessing the ratio of the lower mature (m) and higher precursor (p) bands enabled the estimation of processing of the pre-proteins by mitochondrial processing peptidase (MPP). The processing of the mitochondrial targeting sequence (MTS) was observed for both monomers (Fig. 2A). However, a reduction in import efficiency and expression levels was observed for these C-terminal peptides in a tandem arrangement as compared to the N-terminal peptides (Fig. 2A).

Next, we examined the heteroplasmy shifting capability of tandem configurations of two mtZFN sequential arrangements

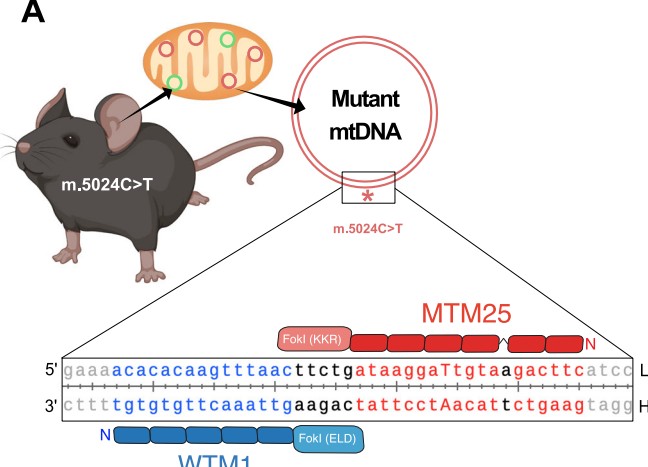

**Figure 1.    Schematic of the m.5024C>T mtZFNs and co-delivery strategy.**

(A) The m.5024C>T mutation, occurring in mt-tRNA[Ala], results in a mismatch in the tRNA acceptor stem, as shown in the clover-leaf representation (top right). This in turn results in secondary structure disruption and lower levels of steady state mt-tRNA[Ala] (Kauppila et al, 2016). To remodel the genotype, an appropriate pair of mtZFN heterodimers was optimized (bottom). Binding sites of both constructs are shown in blue or red, respectively. The mutation site base is capitalized. The 5′ and 3′ ends as well as the heavy (H) and light (L) mtDNA strands are also indicated. (B) The compact nature of mtZFNs enables viral co-encapsidation of both monomers in a single rAAV capsid. Both constructs are encoded in a single ORF by the insertion of a T2A peptide coding sequence (Liu et al, 2017). The residues of the T2A peptide remain appended to the proteins upon dissociation, adding a 17-amino acid chain to the C-terminus of the first protein and a residual proline on the N-terminus of the second one.

(WTM1-T2A-MTM25 or MTM25-T2A-WTM1). To do this, heteroplasmic mouse embryonic fibroblasts (MEFs) harboring ~75% of the m.5024C>T mutation were electroporated with plasmids encoding mtZFNs in various configurations and fluorescent reporters. Upon FACS-based selection, cells were genotyped by pyrosequencing two weeks after transfection (Fig. 2B). Significant heteroplasmy shifts for all mtZFN-treated cells were observed, as compared to control cells transfected with empty plasmids. Constructs administered on separate plasmids showed a more pronounced heteroplasmy shift as compared to both tandem configurations. Importantly, there was a significant difference between the two tandem configurations, which prompted the

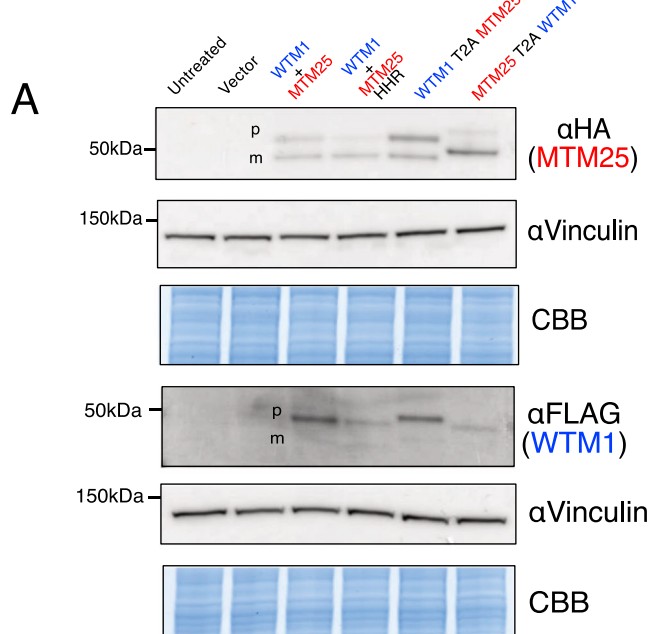

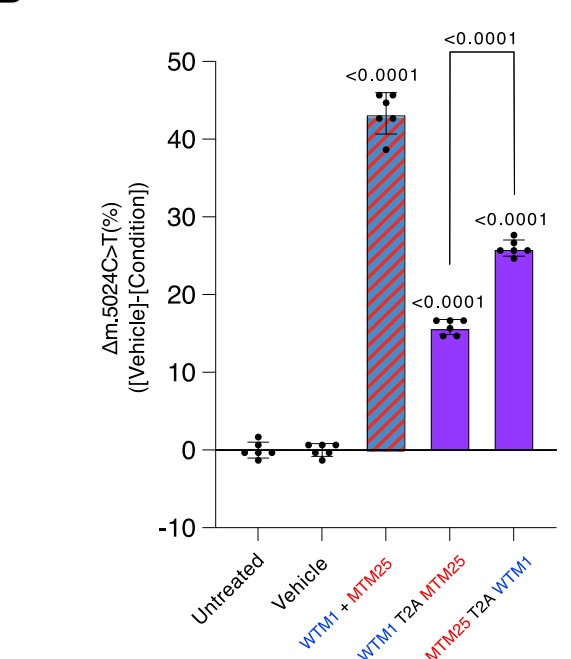

**Figure 2. Effect of sequential arrangement of tandem mtZFN delivery in vitro.**

(A) Western blot analysis of delivery of both heterodimers to HEK293 cells 48 h following transfection. Construct MTM25 was tagged with HA, whereas WTM1 was tagged with FLAG. Heterodimers were either delivered separately (+) or by expression from a single plasmid (T2A) in both sequential arrangements. The two bands are the precursor (p) and mature (m) forms of the peptides. The hammerhead ribozyme (HHR) was used to attenuate expression of the single constructs. All constructs were delivered in equimolar amounts. Coomassie brilliant blue (CBB) and Vinculin are loading controls. (B) Heteroplasmy shifting two weeks after electroporation, as assessed by pyrosequencing. Dual color red and blue denotes separate delivery of constructs on two plasmids. Purple color denotes expression of constructs from a single ORF. Mean ± SEM is displayed. Significance of treated samples was calculated with one-way ANOVA using Dunnett's multiple comparison against 'Vehicle'. Student t-test was used to calculate significance between both orientations of mtZFN in tandem. Each symbol is a biological replicate. Six data points were pooled from two independent biological experiments each having biological triplicates. Source data are available online for this figure.

retention of the MTM25-T2A-WTM1 sequential arrangement for subsequent in vivo experimentation.

## Vector optimization and in vivo comparison of tandem vs. separate mtZFN architecture efficiency

Having determined the functional competence of the tandem mtZFN configuration in mouse cultured cells, we next aimed to generate AAV virions encoding tandem mtZFN MTM25-T2A-WTM1, specific to the mutation site of the m.5024C>T mouse

(Gammage et al, 2018b). While obtaining constructs of the mtZFN transgene within an AAV-compatible DNA plasmid backbone was straightforward, large-scale deletions of the transgene were observed upon generation of AAVs (Appendix Fig. S1). We hypothesized that these deletions were likely due to DNA strand slippage during AAV genome replication, caused by significant homology between the MTM25/WTM1 monomers (93% overall nucleotide sequence identity with identical stretches of over 200 nt long). To address this issue, we exploited DNA codon redundancy by recoding the downstream mtZFN monomer, WTM1, to reduce nucleotide sequence identity and shorten homologous stretches (62% nucleotide sequence identity with no stretch longer than 14 nt) (Appendix supplementary data). This recoded construct yielded AAV genomes of the expected length (Appendix Fig. S1) upon encapsidation, and the resulting viral preparations of MTM25-T2A-WTM1 in the AAV9.45 serotype were used for in vivo experiments.

Next, we aimed to compare the efficacy of the tandem mtZFN configuration with our previous heteroplasmy shifting approach in mouse cardiac tissue using mtZFN monomers encapsidated in separate virions (Gammage et al, 2018b). To this end, a dosage curve of tandem mtZFNs encapsidated within cardiotropic rAAV9.45 was generated, being delivered by tail vein (TV) injection (Pulicherla et al, 2011) into animals harboring between 46% and 78% m.5024C>T mutant heteroplasmy, corresponding to those used for the separate monomer approach previously (Gammage et al, 2018b). For tandem mtZFN, doses are halved with respect to separate monomer doses, as both parts of the nuclease are delivered concurrently to infected cells (Fig. 3A). At 65 days following systemic delivery of rAAV9.45, mice treated with tandem mtZFNs demonstrated significant heteroplasmy shifts away from m.5024C>T in heart tissue, expressed as the change between ear heteroplasmy at weaning and heart heteroplasmy post-mortem, at doses 5E11, 2.5E12, or 5E12 viral genomes (vg)/mouse (Fig. 3B). A dose dependency was observed with tandem mtZFN, producing greater shifts of heteroplasmy than separate monomers at the highest equivalent dose in the curve (5E12 vg/mouse). Treatment with tandem mtZFN demonstrated a lowest effective dose, 5E11 vg/ mouse, which is 10-fold lower than for separate monomers

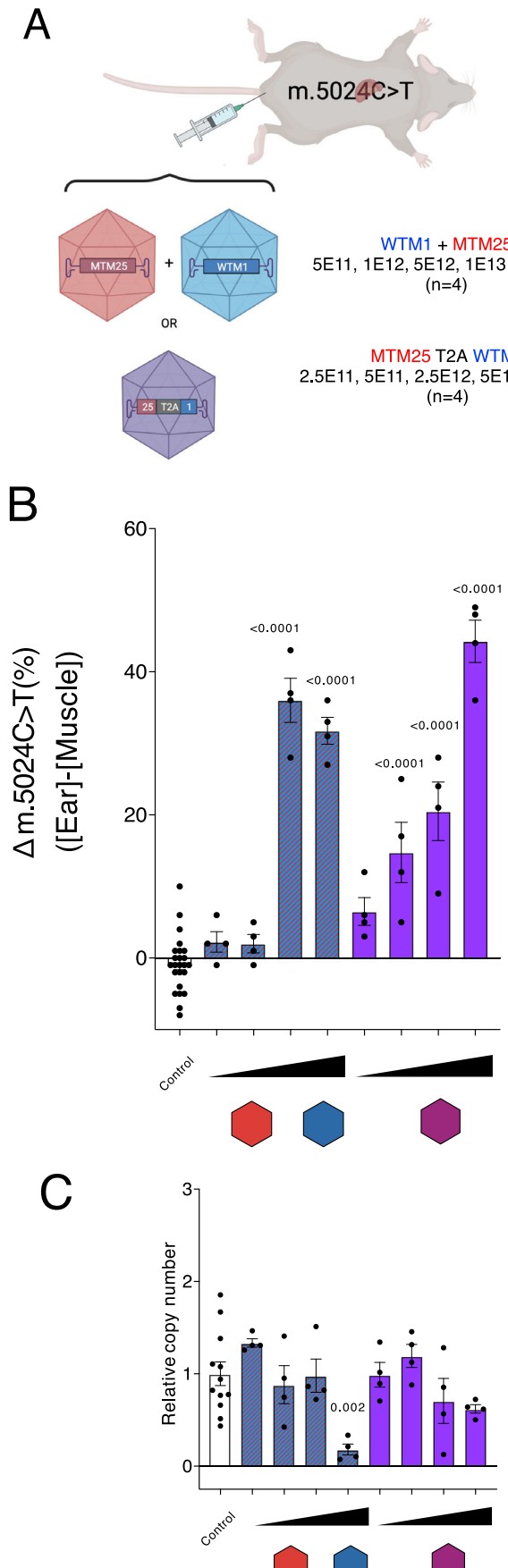

A

m.5024C>T

WTM1 + MTM25
5E11, 1E12, 5E12, 1E13 AAV9.45
(n=4)

OR

MTM25 T2A WTM1
2.5E11, 5E11, 2.5E12, 5E12 AAV9.45
(n=4)

B

<0.0001
<0.0001
<0.0001
<0.0001
<0.0001
<0.0001

Control

C

0.002

Control

**Figure 3. Intravenous delivery of mtZFN using cardiotropic rAAV9.45.**

(A) Schematic representation of experimental design. Tail vein administration of AAV9.45 encapsidated mtZFN, either separately or in the tandem configuration. (B) Heteroplasmy shift measured by pyrosequencing as the difference between initial ear biopsy and final muscle heteroplasmy values 65 days post-rAAV administration. Doses shown in increasing order were 5E11, 1E12, 5E12, and 1E13 vg/mouse for separate virions containing mtZFN monomers (red and blue) and 2.5E11, 5E11, 2.5E12, and 5E12 vg/mouse for the virions containing the tandem mtZFN configuration (purple). NB: Data points for 1E12, 5E12, and 1E13 vg/mouse doses used for separate mtZFN monomers are taken from Gammage et al (2018b) to allow direct comparison with the tandem configuration. Mean ± SEM are displayed, and statistics were performed using one-way ANOVA using Dunnett's comparison to the control. N-numbers were 24 for control samples and 4 biological replicates for each experimental condition. Mean ± SEM are displayed, and statistics were performed using one-way ANOVA using Dunnett's comparison to the control. (C) MtDNA copy number (mtCN) relative to untreated control samples measured by qPCR. Mean ± SEM are displayed, and statistics were performed using one-way ANOVA using Dunnett's comparison to the control. N-numbers were 24 for control samples and 4 biological replicates for each experimental condition. Source data are available online for this figure.

(Fig. 3B). None of the employed doses of tandem mtZFN had statistically significant effects on mtDNA copy number as measured by quantitative PCR (qPCR), although a downward trend in higher dosage conditions was observed (Fig. 3C). In addition, as the tandem mtZFN architecture results in diminished mitochondrial import of WTM1, we sought to determine any effect of tandem mtZFN treatment on nuclear DNA (nDNA) by ultra-deep (>100,000 reads per nucleotide) amplicon resequencing of two loci within nDNA that have high homology (>90% sequence identity) to the MTM25/WTM1 binding site in mtDNA (Fig. 2). No changes to the level of insertion/deletion (InDel) formation in total heart DNA samples between vehicle and tandem mtZFN AAV-injected animals at 5E11 vg/mouse were observed (Appendix Fig. S2). This suggests that the tandem mtZFN nuclear DNA off-target effect profile is negligible, similarly to that of controls and separate monomers, analyzed previously (Gammage et al, 2018b). Taken together, these results show that tandem mtZFN, delivered systemically, demonstrate significant heteroplasmy shifts in mouse cardiac tissue, with greater efficacy and lower effective doses compared to mtZFN monomers delivered separately, without significant off-target effects on nuclear DNA or mtDNA.

## Intramuscular administration of tandem mtZFN by AAV9 results in localized heteroplasmy shift rescue of molecular phenotype

Next, we intended to test the novel tandem mtZFN configuration to address mtDNA heteroplasmy shifting in skeletal muscles, which is relevant for a large group of primary mtDNA diseases (Chinnery, 2021). To this end, local intramuscular injections as a myotropic treatment of the m.5024C>T mouse were performed. Such an experimental setup allowed for an internal negative control by administering a mock treatment to a different muscle within the same animal. Tandem or separate rAAV9 mtZFN treatments were administered to the right quadriceps muscles of m.5024C>T mice harboring between 54% and 64% m.5024T heteroplasmy. Treatments were administered to contain equimolar amounts of mtZFNs, with 1E12 vg of the tandem condition versus 2E12 vg

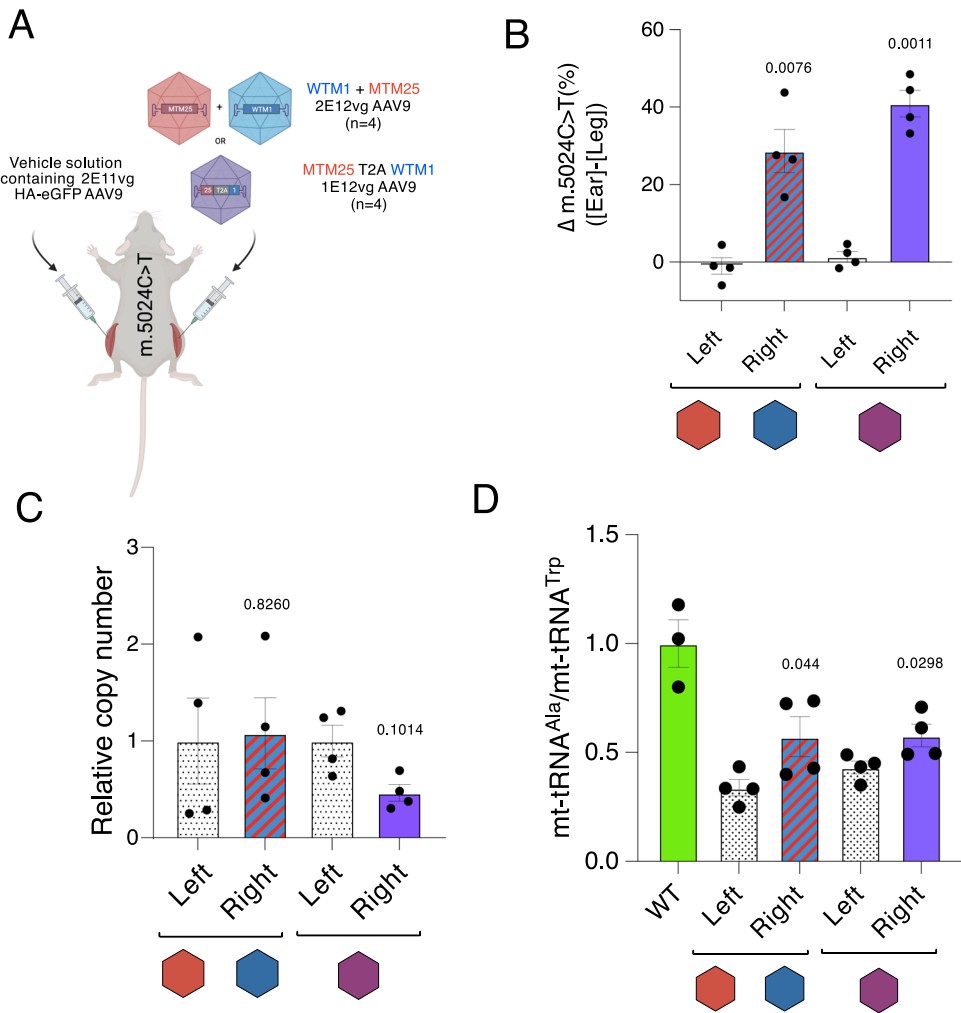

**Figure 4. Intramuscular administration of mtZFN.**

(A) Schematic representation of experimental design. Left legs were injected with rAAV buffering solution containing HA-eGFP mock construct as a positive marker for injection. Right legs were injected with rAAVs encoding mtZFNs, either in separate viral particles or encapsulated in tandem, diluted with control solution for left leg. Treatments were delivered in equimolar amounts, with four biological replicates (mice) per condition. Statistics were performed using paired t-test. (B) Heteroplasmy shift measured by pyrosequencing as the difference between initial ear biopsy and final muscle heteroplasmy values 65 days post rAAV administration. Mean ± SEM are displayed and statistics for both experiments were performed by paired t-test. N-number was four for both experimental conditions. (C) Relative mtCN in both left and right legs assessed by qPCR. Mean ± SEM are displayed and statistics for both experiments were performed by paired t-test. $p = 0.826$ for WTM1 + MTM25 and $p = 0.1014$ for MTM25-T2A-WTM1. (D) Pooled and averaged ratios from digitally quantified and normalized northern blot. Statistics were performed using paired t-test. $p = 0.049$ for WTM1 + MTM25 and $p = 0.0298$ for MTM25-T2A-WTM1. N-number was four for both experimental conditions. Source data are available online for this figure.

of the separately administered viruses. Furthermore, 2E11 vg of rAAV9 encoding HA-tagged eGFP were included as a proxy for injection delivery (Fig. 4A). Muscles were harvested 65 days following administration and viral delivery to the right leg target muscle was confirmed by western blotting (Appendix Fig. S3). Heteroplasmy shifting in the treated muscles was assessed by pyrosequencing, comparing the mutation load in the ear biopsy prior to the start of the experiment with the muscle heteroplasmy values. Highly significant reduction in heteroplasmy in the right legs versus the control left legs of both treatment groups was detected, with separate capsid delivery, achieving a ~28% reduction in mutation load, while nearly 40% reduction observed for the tandem administration (Fig. 4B). The alterations in mtDNA copy

number (mtCN) were also assessed by comparing the relative mtCN between the left and right legs, with no significant changes in mtCN being observed, despite increasing and decreasing trends for the separate and tandem conditions, respectively (Fig. 4C).

As the m.5024C>T mutation results in reduced steady-state levels of mt-tRNA^Ala, this feature was used to assess phenotypic rescue after the mtZFN treatment. To this end, the steady-state level of mt-tRNA^Ala was analyzed by northern blotting in wild-type muscle samples, along with samples from the left and right legs of the local treatment experiment (Fig. 4D; Appendix Fig. S4). Significant increases in the steady state levels of mt-tRNA^Ala compared to mock-treated left legs were detected, with approximately a 1.7-fold increase and a 1.3-fold increase in the separate

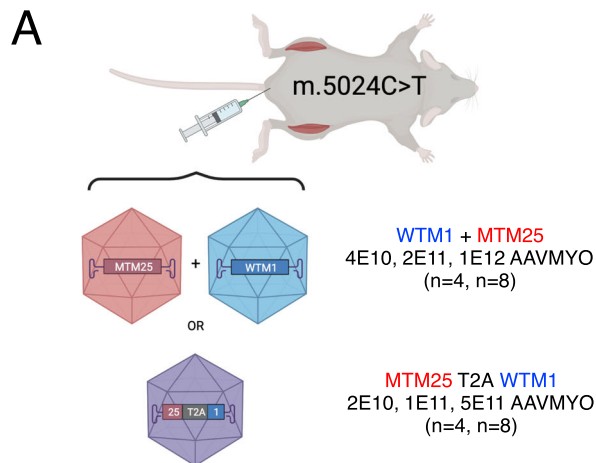

A

m.5024C>T

WTM1 + MTM25
4E10, 2E11, 1E12 AAVMYO
(n=4, n=8)

OR

MTM25 T2A WTM1
2E10, 1E11, 5E11 AAVMYO
(n=4, n=8)

**Figure 5. Intravenous delivery of mtZFNs using myotropic rAAV.**

(**A**) Schematic representation of experimental design. Tail vein administration of AAVMYO encapsidated mtZFNs, either separately or in tandem. (**B**) Heteroplasmy shift measured by pyrosequencing as the difference between initial ear biopsy and final muscle heteroplasmy values 65 days post-rAAV administration. Doses shown in increasing order are 4E10 (4 biological replicates), 2E11 (4 biological replicates) and 1E12 (8 biological replicates) vg/mouse for both virions containing mtZFN monomers (red and blue) and 2E10 (4 biological replicates), 1E11 (4 biological replicates) and 5E11 (8 biological replicates) vg/mouse for the virions containing the tandem mtZFN configuration, with 4 control animals. Mean ± SEM are displayed, and statistics were performed using one-way ANOVA using Dunnett's comparison to the control (**C**) mtCN relative to untreated control samples were measured by qPCR. Mean ± SEM are displayed, and statistics were performed using one-way ANOVA using Dunnett's comparison to the control. Doses shown in increasing order are 4E10 (4 biological replicates), 2E11 (4 biological replicates), and 1E12 (8 biological replicates) vg/mouse for both virions containing mtZFN monomers (red and blue) and 2E10 (4 biological replicates), 1E11 (4 biological replicates), and 5E11 (8 biological replicates) vg/mouse for the virions containing the tandem mtZFN configuration, with 4 control animals. Source data are available online for this figure.

B

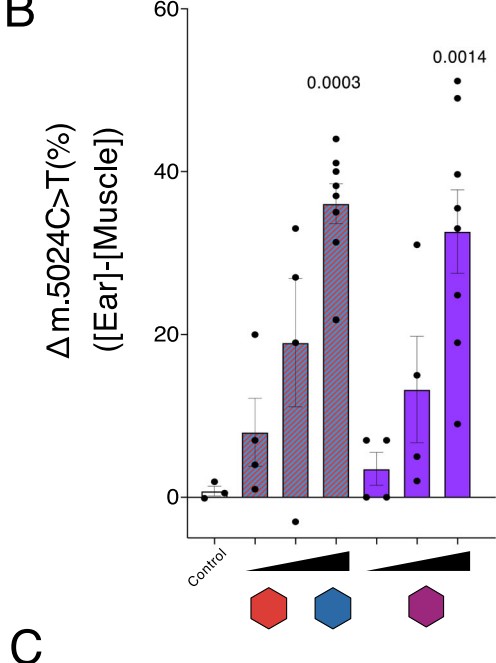

C

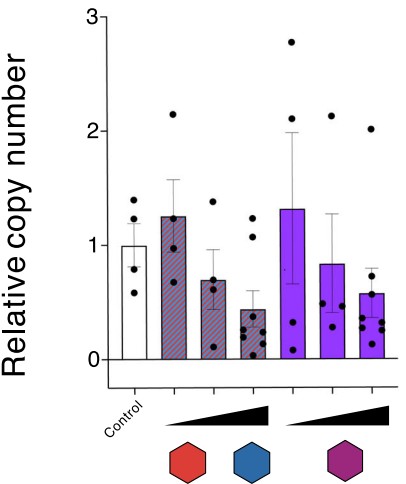

and tandem conditions, respectively. The rescue amounts, however, were lower than the steady-state levels found in the wild-type mice of the same background strain, with both rescues only restoring ~57% of the steady-state amounts of mt-tRNA (Fig. 4D), in line with the incomplete mutant rescue observed post mtZFN treatment. Taken together, these results demonstrate that the novel tandem mtZFN configuration significantly reduces mtDNA heteroplasmy and achieves phenotypic rescue in skeletal muscles of m.5024C>T mice, with enhanced efficacy compared to separate mtZFNs.

## Systemic administration of mtZFNs by AAVMYO results in reduction of mutation load across the musculature

Local intramuscular administration proved to be successful in remediating mutation load and downstream molecular phenotype as well as demonstrating the effectiveness of the novel tandem delivery method applied in vivo to a clinically relevant tissue. Despite this, the lack of wide distribution coupled with the current impracticality of re-administering rAAV treatments hinder the clinical translatability of intramuscular delivery. Therefore, our next objective was to deliver the treatment more broadly to muscle using a novel myotropic AAV9-derived vector, termed AAVMYO (Weinmann et al, 2020). To this end, mice harboring m.5024C>T (between 50% and 70%) were administered three doses, with 5-fold differences between dose levels, of either separate of tandem m.5024C>T-specific mtZFN architecture encapsulated within AAVMYO. Viruses were administered in an equimolar fashion; hence doses were twice as high for the separate, non-tandem mtZFN viral administrations. Four mice were treated per condition, except for the highest dosage condition where four additional mice were injected, as this dose level served as a benchmark between two independent viral administrations (Fig. 5A). The quadriceps muscle was analyzed for these experiments for consistency with the intramuscular experiment. Heteroplasmy shifting in the AAVMYO-mtZFN-treated muscles was assessed by pyrosequencing, comparing the mutation load in the ear biopsy prior to the start of the experiment with the final muscle heteroplasmy values.

All dose levels showed decreases in m.5024C>T heteroplasmy with the highest dose showing strong significance, achieving heteroplasmy shifts of 36% and 32% in the separate and tandem conditions, respectively (Fig. 5B). mtCN variation was assessed and showed no significant changes, however, a progressive decrease proportional to the increase in viral dosage was observed (Fig. 5C). These findings show that the new tandem mtZFN configuration can effectively reduce mtDNA heteroplasmy in the skeletal muscles of m.5024C>T mice upon systemic delivery.

### AAV-mediated mtZFN delivery results in minor immune and inflammatory responses which are further diminished by tandem delivery

Next, we aimed to assess, the safety profile of systemic mtZFN treatment by measuring transcript levels of inflammatory and immune markers, to investigate how varying viral dosages and routes of administration of mtZFNs impact innate immunity and inflammation. In particular, since the induction of double-strand breaks using mitoTALENs has been linked with triggering innate immunity in vitro (Tigano et al, 2021), we included key transcript targets belonging to the anti-viral type I interferon (IFN) response, *Ifnb1*, *Cxcl10*, *Ddx58*, and *Isg20* and the pro-inflammatory cytokines, *Il1b* and *Tnf*. For intramuscular administration, after confirming genotypic and molecular phenotypic rescue following treatment (Fig. 4), the levels of these transcripts were assessed by RT-qPCR and compared between the mock-treated left and experimental right legs of the mice (Appendix Fig. S5). No significant changes between the left and right legs of the selected targets were found, consistent with an absence of innate immune or inflammatory response within the panel of targets chosen. We then investigated the dose dependency of the immune and inflammatory responses following systemic AAVMYO administration. The aforementioned transcript levels were compared between untreated age and heteroplasmy matched mice and the three AAV doses (Fig. 5A) for both mtZFN treatment conditions (Fig. 6). Only the highest dose of mtZFN administered separately (1E12) displayed a small, but significant increase in pro-inflammatory *Il1b* and *Isg20* (Fig. 6A,F) expression with a similar but non-significant trend in expression seen in *Tnf* (Fig. 6B). The equivalent dose for the tandem architecture (5E11) did not elicit any detectable response. Likewise, the highest dose of mtZFN administered separately (1E12) only led to a small but significant increase in the type I IFN response gene, *Isg20*, but none of the other targets, with the equivalent dose for the tandem architecture (5E11) not triggering any significant response. In contrast, significant reductions in RIG-I (*Ddx58*) were observed across lower dose levels, namely at the 4E10 and 2E11 dose levels for separate delivery and the 2E10 dose for tandem delivery, with reductions in some conditions nearing 20-fold (Fig. 6E). Despite trending upwards at higher doses of AAV, no significant upregulation in inflammatory markers was observed for either *Tnfa*, *Ifnb1*, or *Cxcl10* (Fig. 6B–D). In addition, local intramuscular delivery of separate and tandem mtZFNs led to no significant pro-inflammatory or type I IFN response (Appendix Fig. S5) when compared to the matching internal control. Overall, the results indicate a minor in vivo immune response to systemically administered mtZFNs delivered using two separate AAVs. However, the decreased viral dose associated with tandem delivery yields a significantly decreased immune response to treatment.

### Dose titration of AAVMYO enables genotypic and phenotypic rescue at low levels of viral transduction

Given that both separate and tandem administration of AAVMYO resulted in comparable performances in terms of gene therapy (Fig. 5), but that only the separate administration showed statistically significant upregulation in *Il1b* and *Isg20* immune markers (Fig. 6), further investigations focused on the tandem arrangement. In these experiments we aimed to establish the lowest effective dose that would rescue the molecular phenotype and to examine whether dose escalation enhances heteroplasmy shifts. To this end, in addition to the 2E10, 1E11, and 5E11 doses (Fig. 5A), 2.5E11 and 2.5E12 vg/mouse AAVMYO-mtZFN were administered to mice harboring the m.5024C>T mutation (between 50% and 70%) for a period of 65 days (Fig. 7A). Heteroplasmy analysis, revealed a "plateau" effect at the highest dose level, where an increase in the AAV titer did not seem to further enhance the magnitude of the heteroplasmy shift (Fig. 7B). While 2.5E11 reduced the levels of m.5024C>T by ~20%, statistical significance was not reached (Fig. 7A), identifying 5E11 vg/mouse as the lowest effective dose. To confirm that the treatments with AAVMYO restored the steady-state levels of mt-tRNA$^{Ala}$, northern blot analysis was performed at the 5E11 vg dose level. Significant restoration of the steady-state levels of mt-tRNA$^{Ala}$ compared to the vehicle treated heteroplasmy matched control was observed (Fig. 7C). Investigation of mtCN by qPCR showed a progressive dose-dependent decrease, with a substantial reduction at the highest dose level approaching statistical significance upon analysis (Fig. 7D). To further assess the safety of treatment with mtZFNs delivered by AAVMYO, nDNA off-target analysis was performed at the tandem 2.5E11 and 5E11 treatment conditions as well as the separate 1E12 condition by next-generation sequencing. By aligning the target mitochondrial locus to the murine C57BL/6 reference genome, two regions of near-perfect homology were identified on chromosomes 2 and 5, respectively. These regions were amplified in the control and treated mice and screened for the presence of insertions or deletions (InDels) to assess aberrant nuclease activity in the nucleus. No significant increase in InDels was observed when comparing the treated samples to the control mice, suggesting negligible mtZFN presence and activity in the nucleus (Appendix Fig. S6). Taken together, we established the lowest effective and safe dose for AAVMYO at 5E11 vg/mouse for molecular phenotype rescue of the m.5024C>T mutation in skeletal muscle.

## Discussion

This study aimed to optimize single-AAV-delivered mtZFNs for gene therapy of mtDNA mutations using the m.5024C>T mouse model. It evaluated tandem versus separate mtZFN monomer administrations for inducing heteroplasmy shifts and rescuing molecular phenotypes in cardiac and skeletal muscle using AAV9.45, AAV9, and AAVMYO vectors. In addition, it sought to identify the lowest effective AAVMYO dose for systemic injections to achieve significant genotypic and phenotypic rescue in skeletal muscle while minimizing immune responses.

The initial in vitro experiments in cultured cells showed that tandem administration of mtZFNs using the T2A "self-cleaving"

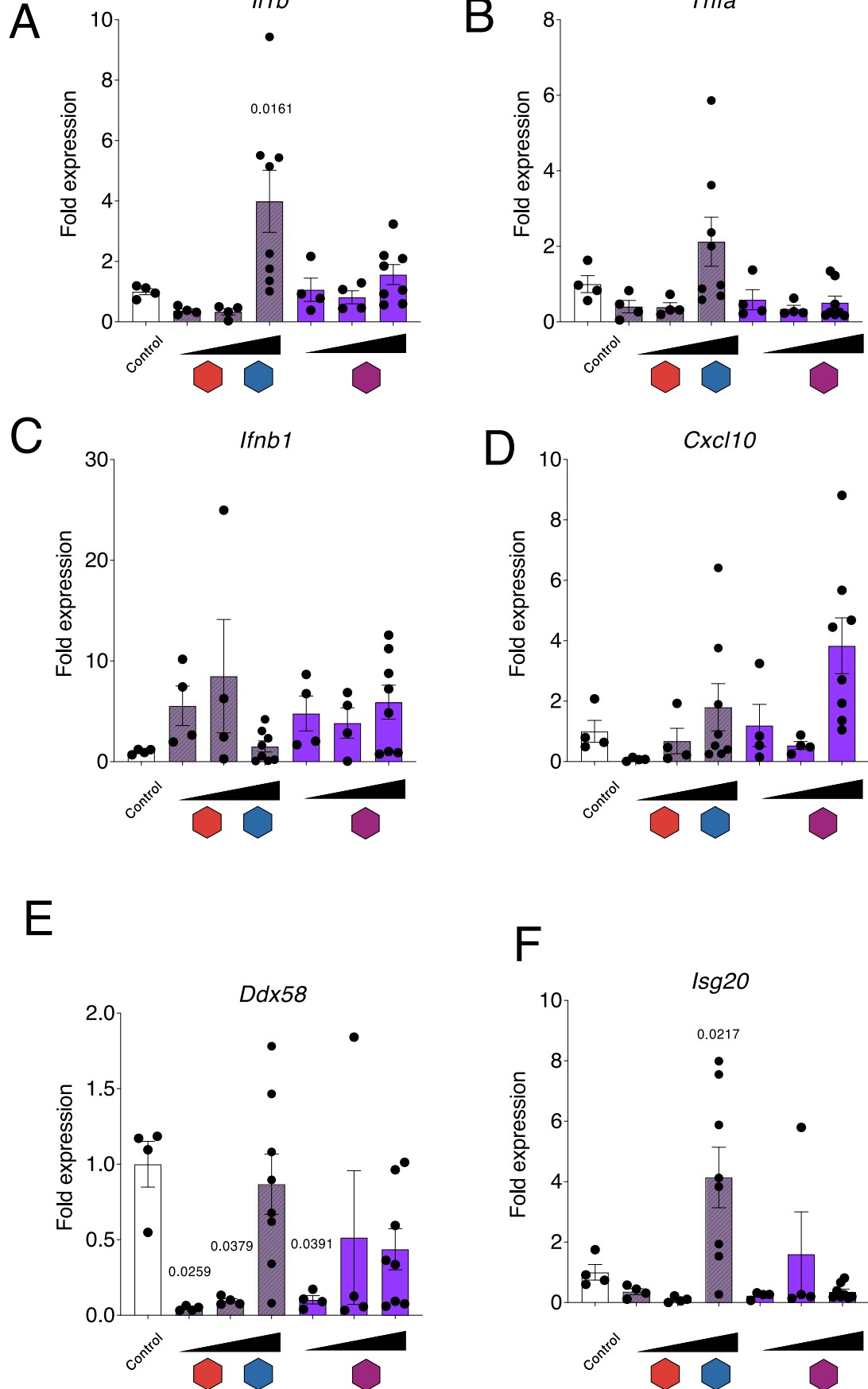

**Figure 6.  Innate immune responses to muscle gene therapy.**

RT-qPCR measurements of transcripts related to pro-inflammatory cytokines and the type I interferon response obtained for both separate and tandem administrations. The following transcripts were analyzed: (A) *Il1b*, (B) *Tnfa*, (C) *Ifnb1*, (D) *Cxcl10*, (E) *Ddx58*, (F) *Isg20*. The doses were 4E10 (4 biological replicates), 2E11 (4 biological replicates), and 1E12 (8 biological replicates), vg/mouse for virions containing mtZFN monomers (red and blue), and 2E10 (4 biological replicates), 1E11 (4 biological replicates), and 5E11 (8 biological replicates) vg/mouse for virions containing the tandem mtZFN configuration, with 4 control animals. Samples were collected 65 days post-rAAV administration. Samples were performed in technical quadruplicate and fold change calculated compared to age and heteroplasmy matched control mice. Bar plot shows mean ± SEM. Statistical analysis was conducted using one-way ANOVA, with results compared to the control group utilizing Dunnett's test. Source data are available online for this figure.

peptide is a successful strategy regardless of the sequential arrangement of the two constructs. It was interesting that equimolar delivery of mtZFN monomers as separate or tandem constructs, led to differences in their import into mitochondria and their ability to shift heteroplasmy. One explanation is that protein scars after separation at T2A may alter the properties of mtZFN. The presence of proline at the N-terminal of the MTS in the distal protein could impede mitochondrial import, or the 17 amino acids present at the C-terminus of the proximal monomer might interfere with FokI's nucleolytic activity (Fig. 1B). Furthermore, the longer ORF when using the tandem arrangement reduces the expression levels of the distal protein, affecting the overall levels of the heterodimer and possibly the overall nucleolytic activity of the pair. This effect could advantageous therapeutically, as we have previously shown that a reduction in nuclease expression levels using HHR can minimize mtCN depletion and enhances specificity (Gammage et al, 2016). In the in vitro experiments, the MTM25-T2A-WTM1 arrangement resulted in a greater heteroplasmy shift as compared to WTM1-T2A-MTM25 (Fig. 2B), perhaps due to a greater abundance of the MTM25 construct in its processed form as observed by western blotting (Fig. 2A). For other mtZFN pairings, each arrangement would need to be tested individually due to variable binding kinetics of each different ZF array and its processing efficiency by the mitochondrial translocation pathway. This need for individualized testing is highlighted by the consistently lower levels of mitochondrial uptake observed for the WTM1 construct in all conditions (Fig. 2A), despite WTM1 being a shorter peptide than MTM25, as typically shorter ZF peptides are imported into mitochondria more efficiently (Minczuk et al, 2006). Further development of the technology for single AAV delivery of mtZFNs could involve incorporating an internal MPP site, however, it would be crucial to verify MPP processing efficiency far (>500 aa in case of mtZFNs) from the N-terminus.

We next demonstrated that re-coding of the AAV-compatible transgene avoids recombination events during AAV replication (Appendix Fig. S1), meaning tandem mtZFNs can be packaged into AAV virions. Both local and systemic administration experiments yielded similar maximal heteroplasmy shifts of ~40%, with a clear dose dependency in the systemic administration experiments (Figs. 3B, 5B and 7B). Importantly, the tandem mtZFN architecture administered by AAV9.45 allowed 10-fold lower effective doses of mtZFNs, when compared with previously published data (Gammage et al, 2018b), likely due to co-transduction dynamics at the single cell level across tissues. For AAV9.45 doses greater or equal to 5E11 reached significance (Fig. 3), with the same being observed for 5E11 AAVMYO (Fig. 7). However, the magnitude of heteroplasmy shift for AAVMYO 5E11 was approximately double that of AAV9.45 5E11 (Figs. 3B and 7B).

Tandem mtZFNs administered by AAV9.45 had lower copy number depletion at the higher end of the dosage range as compared to separate monomers (Fig. 3). However, the data on mtCN depletion for systemic administration of AAVMYO is more confounding, with both mtZFN arrangements yielding highly variable results, not reaching significance by analysis with one-way ANOVA. Despite this, a clear trend in mtCN change could be observed, particularly in the dose-dependent AAVMYO experiment, where the trend of mtCN is negatively correlated with dose, with the highest dose nearing significance ($p = 0.067$) (Fig. 7D). Interpreting the heteroplasmy and mtCN data for each dose suggests that optimal AAVMYO dose levels exist where significant heteroplasmy shifts can be achieved whilst minimizing mtCN depletion, similar to the results for cardiac muscle upon systemic administration of AAV9.45.

Local administration of mtZFNs to the right leg only was advantageous when analyzing the molecular phenotypic rescue of the steady state levels of mt-tRNA[Ala] (Fig. 4D). Normalization of the relative abundance of mt-tRNA[Ala] in the right leg versus the left to mt-tRNA[Trp] gave a rigorous quantification. Subsequent administrations of AAVMYO were also accompanied with improvements in mt-tRNA[Ala] abundance (Fig. 7C). While rescue of the mitochondrial molecular phenotype in this mouse model has now been confirmed by multiple studies with various technologies (Gammage et al, 2018b; Bacman et al, 2018; Zekonyte et al, 2021), whether molecular rescues on the level of transcripts permeate to higher phenotypes in vivo remains an open question. This question could benefit from mouse models harboring more pronounced disease phenotypes resulting from underlying mitochondrial mutations (Burr et al, 2023; Silva-Pinheiro et al, 2023).

The safety profiling of immune response markers for both local AAV9 and systemic AAVMYO administration was measured. No significant change in expression levels of *Ifnb1*, *Cxcl10*, *Ddx58*, *Il1b*, or *Tnf* were detected for local AAV9 administrations (Appendix Fig. S5). In an equivalent set of experiments performed on the AAVMYO treated samples (Fig. 6), small but significant increases in *Il1b* and *Isg20* were detected for the separate capsid administration at the highest dose (1E12) compared to the age and heteroplasmy matched untreated mice and to mice treated with tandem mtZFN, suggesting a correlation with viral load. Interestingly, a reduction of the gene expression of Ddx58 coding for the RIG-I protein was detected at lower doses of AAV. It is conceivable that the low immunogenicity of rAAVs is partially the result of immunosuppressive activity, which could explain some of the findings. It has been observed that certain engineered serotypes of rAAV result in diminished levels of innate immunity and inflammatory markers, including *Ddx58*, when transduced into

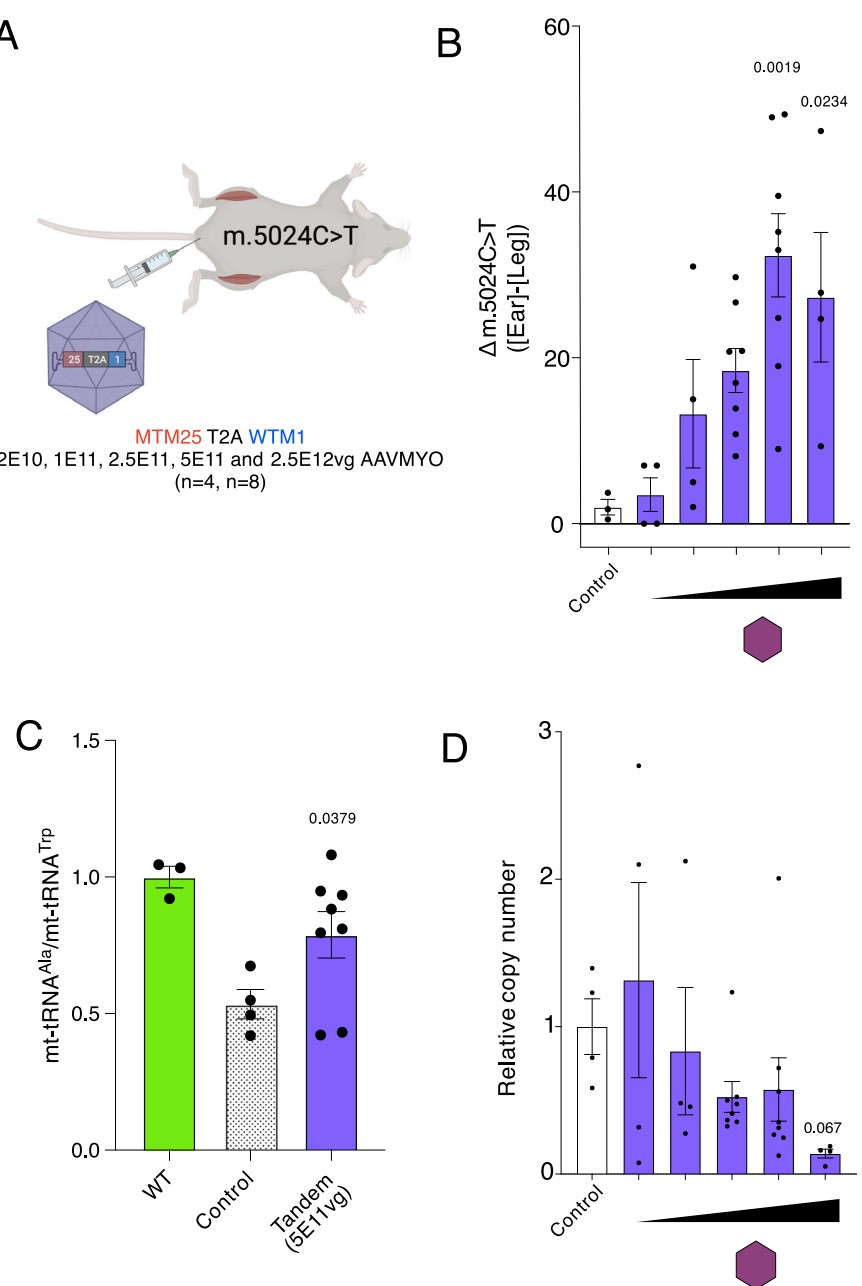

**Figure 7. Dose titration of AAVMYO encoding tandem mtZFN.**

(A) Schematic representation of experimental design. (B) Heteroplasmy shift measured by pyrosequencing as the difference between initial ear biopsy and final muscle heteroplasmy values 65 days post rAAV administration. Doses shown in increasing order 2E10 (4 biological replicates), 1E11 (4 biological replicates), 2.5E11 (8 biological replicates), 5E11 (8 biological replicates), and 2.5E12 (4 biological replicates) vg/mouse of each virion containing the tandem mtZFN configuration, with 4 control animals. Mean ± SEM are displayed, and statistics were performed by one-way ANOVA with Dunnett comparison to control. (C) Pooled and averaged ratios from digitally quantified and normalized northern blot. Statistics were performed using t-test between control and mice treated with 5E11 (8 biological replicates) of the tandem virus. 3 C57BL/6J wild-type mice and four control mice were analyzed. (D) mtCN relative to untreated control samples measured by qPCR. Mean ± SEM are displayed, and statistics were performed using one-way ANOVA using Dunnett's comparison to the control. Doses shown in increasing order 2E10 (4 biological replicates), 1E11 (4 biological replicates), 2.5E11 (8 biological replicates), 5E11 (8 biological replicates), and 2.5E12 (4 biological replicates) vg/mouse of each virion containing the tandem mtZFN configuration, with 4 control animals. Source data are available online for this figure.

human cells (Bentler et al, 2023). Another noteworthy observation is that unlike in vitro (Tigano et al, 2021), mtDNA double-strand breaks in vivo did not upregulate a type I IFN immune response to the same extent. Overall, the data obtained suggest no major

immune or inflammatory response to mtZFN treatments administered by rAAV.

Efficient delivery of exogenous genetic material in humans by AAV, whether focal, targeted, or systemic, requires administration

of significant quantities of viral material. The majority of mitochondrial diseases affect multiple organ systems, likely necessitating organ-wide transduction for therapeutic benefit. While our previous work served as proof-of-principle for mtZFN-mediated heteroplasmy shifting and molecular phenotype rescue across an entire organ, it used AAV titers that were not scalable for human use (Gammage et al, 2018b). Therefore, determining the lowest effective doses for mtZFNs has become a major focus in the path towards clinical application. The developments described here enhance mtZFN efficacy and allows for lower AAV titers, representing a further step toward delivering AAV-mtZFN for human gene therapy targeting heteroplasmic mtDNA disease (Appendix Fig. S7). Ultimately, while both local and systemic administration experiments yielded highly successful genotypic and phenotypic outcomes, systemic administration offers more promising prospects for future clinical applications due to its wide distribution and simplicity of administration. In addition, the continual development of rAAV capsids and other delivery vectors such as nanoparticles will further reduce dosage requirements and enable greater tissue specificity, bringing mitochondrial gene therapy closer to the clinic.

# Methods

**Reagents and tools table**

| Reagent/Resource | Reference or Source | Identifier or Catalog Number |
| --- | --- | --- |
| **Experimental models** | | |
| C57BL/6j-tRNAAla mice | Kauppila et al, 2016 | N/A |
| MEF derived from tRNAAla | Gammage et al, 2018b | N/A |
| HEK293T | ATCC | CRL-3216™ |
| **Recombinant DNA** | | |
| GeneArt™ synthesized constructs | Thermo Fisher Scientific | N/A |
| rAAV2-CMV plasmids | Gammage et al, 2018b | N/A |
| **Antibodies** | | |
| HA | Roche | 11867423001 |
| FLAG | Sigma-Aldrich | F1804 |
| Vinculin | Thermo Fisher | MA5-11690 |
| Anti-Rat IgG (HRP) | Santa Cruz | SC2065 |
| Anti-mouse IgG (HRP) | Promega | W4021 |
| **Oligonucleotides and other sequence-based reagents** | | |
| Oligonucleotides | This study | Table provided separately |
| **Chemicals, Enzymes, and other reagents** | | |
| cOmplete™ Protease Inhibitor Cocktail | Roche | 4693132001 |
| BCA Protein Assay Kit | Thermo Fisher Scientific | 23250 |
| MES SDS Running Buffer | Invitrogen™ | NP0002 |

| Reagent/Resource | Reference or Source | Identifier or Catalog Number |
| --- | --- | --- |
| NuPAGE™ Sample Reducing Agent | Invitrogen™ | NP0009 |
| NuPAGE™ LDS Sample Buffer | Invitrogen™ | NP0007 |
| TRIzol™ reagent | Thermo Fisher Scientific | 15596026 |
| NexteraXT DNA Library Preparation Kit | Illumina | FC-131-1096 |
| EcoRI | New England Biolabs | R3101L |
| BamHI | New England Biolabs | R3136L |
| EagI | New England Biolabs | R3505L |
| BglII | New England Biolabs | R0144L |
| T4 DNA Ligase | New England Biolabs | M0202T |
| High glucose Dulbecco's modified Eagle media (DMEM) | Thermo Fisher Scientific | 11965092 |
| FuGENE® HD | Promega | E2311 |
| α[32P]-UTP | Hartman | SRP210 |
| Kaleidoscope™ | Bio-Rad | 1610375 |
| Amersham ECL™ Western Blotting Detection Reagents | GE Healthcare | RPN2105 |
| Pyromark® PCR kit | QIAGEN | 978703 |
| Pyromark® magnetic beads | QIAGEN | 974203 |
| PowerUp SYBR Green Master Mix | Applied Biosystems | A25742 |
| Omniscript Reverse Transcription | QIAGEN | 205111 |
| Oligo(dT)20 Primer | Invitrogen™ | 18418020 |
| RNAsin® RNase inhibitor | Promega | N2515 |
| Phusion® DNA polymerase | New England Biolabs | M0536S |
| **Software** | | |
| QuantStudio™ 3 Real-Time PCR System | Thermo Fisher Scientific | N/A |
| Typhoon phosphor imaging system | GE Healthcare | N/A |
| GraphPad Prism | GraphPad Software LLC | N/A |
| Pyromark Q48 Autoprep | QIAGEN | N/A |
| **Other** | | |
| gentleMACS™ Dissociator | Miltenyi Biotec | 130-093-236 |
| One Shot™ TOP10 Chemically Competent E. coli | Thermo Fisher Scientific | C404010 |
| Nucleofector II | Lonza | N/A |
| MEF1 kit | Lonza | VPD-1004 |
| SDS-PAGE 4–12% bis–tris gels | Life Technologies | NP0321BOX |
| iBlot 2 Dry Blotting System | Thermo Fisher Scientific | IB21001 |
| MiSeq/HiSeq/NovaSeq | Illumina | N/A |

| Reagent/Resource | Reference or Source | Identifier or Catalog Number |
|---|---|---|
| iBlot 2 Transfer Cell | Life Technologies | IB24001 |
| Hybond-N+ Nylon Membrane | Cytiva | RPN303B |
| NanoDrop™ Spectrophotometer | Thermo Fisher Scientific | N/A |
| Amicon Ultra-0.5 Centrifugal Filter Units | Merck | UFC5100 |
| Amersham™ Imager 600 | GE Healthcare | N/A |
| Pyromark® Q48 disks | QIAGEN | 974901 |
| Amicon Ultra-0.5 Centrifugal Filter Units | Merck | UFC5100 |

## Methods and protocols

### Constructs, plasmids, and viral vectors

Designed constructs were synthesized by ThermoFisher Scientific Inc. using the GeneArt™ synthesis portal. Zinc finger arrays were cloned into plasmid backbones containing second-generation mtZFNs flanked single cutting restriction sites *EcoRI* and *BamHI* (Gammage et al, 2014). Furthermore, these plasmids contain fluorescent reporters eGFP and mCherry, respectively, for each heterodimer, allowing for subsequent fluorescence-activated cell sorting (FACS). Cloning took place as described previously (Nash and Minczuk, 2023). In brief: GeneArt plasmids containing zinc finger arrays as well as the corresponding backbones were digested using *EcoRI-HF* (NEB - R3101L) and *BamHI-HF* (NEB - R3136L) enzymes for 1 h at 37 °C and subsequently resolved by 1% agarose gel electrophoresis. Fragments were excised and purified using QIAquick Gel Extraction Kit (QIAGEN - 28704) before being ligated in a 1 to 5 backbone to insert molar ratio using a 20 μL T4 DNA Ligase reaction (NEB - M0202T) for 20 min at 25 °C. Competent cells (TF - C404010) were transformed with 5 μL of ligation reaction as per the manufacturer's protocol, with 200 μL of resulting outgrowth being plated on 10 cm LB Agar plates and left to incubate overnight at 37 °C.

Plasmid construction of mtZFNs intended for AAV production was achieved by PCR amplification of MTM25(+)-HHR, WTM1 (-)-HHR, and MTM25(+)-T2A-WTM1(-) transgenes, incorporating 5' *Eag*I and 3' *Bgl*II sites. These products were then cloned into rAAV2-CMV between 5' *Eag*I and 3' *BamH*I sites. The FLAG epitope tag of WTM1(-) was replaced with a hemagglutinin (HA) tag at the same position in the WTM1(-) open reading frame by PCR. The 3K19 hammerhead ribozyme (HHR) sequence (Beilstein et al, 2015) was incorporated into separate monomer mtZFN constructs as described above (Nash and Minczuk, 2023). The resulting plasmids were used to generate recombinant AAV2/9.45-CMV-MTM25, AAV2/9.45-CMV-WTM1, and AAV2/9.45-CMV-MTM25_T2A_WTM1 viral particles at the UNC Gene Therapy Center, Vector Core Facility (Chapel Hill, NC, USA). The same vectors were used to produce the viruses of AAV2/9 serotype for local intramuscular injections at UNC Gene Therapy Center, Vector Core Facility (Chapel Hill, NC, USA) and of AAVMYO serotype for systemic injections. All viruses were adjusted to correct experimental concentration using Amicon Ultra-0.5 Centrifugal Filter Units (Merck – UFC5100) and diluted in rAAV dilution buffer [1X PBS, 350 mM NaCl, 5% D-Sorbitol].

### Cell lines and culture conditions

Both HEK293T and mt-tRNA[Ala] MEFs were cultured in high glucose Dulbecco's modified Eagle media (DMEM) supplemented with 10% (v/v) fetal bovine serum (FBS), 1X non-essential amino acids, 50 μg/ml penicillin, 50 μg/ml streptomycin, and 50 μg/ml uridine. Cell cultures were maintained at 37 °C under 5% $CO_2$ atmosphere. Cell cultures were regularly tested for mycoplasma contamination.

### Transfection of HEK293T cells

300k cells were seeded into 6-well plates a day prior transfection to ensure they were 70–80% confluent on the day of transfection. On the day of transfection, for each microgram of plasmid DNA, 2.5 μL of FuGENE® HD reagent (Promega - E2311) was used, adhering to a 2.5:1 FuGENE® HD to DNA ratio. The reagent and DNA were mixed gently in serum-free Opti-MEM™ media and allowed to incubate for 10–15 min at room temperature to form complexes. The DNA-FuGENE® HD mixture was then added dropwise to the cells. Post transfection, cells were incubated under standard conditions for 48 h.

### Electroporation and FACS of MEFs

Electroporation on mouse embryonic fibroblasts (MEFs) was performed using Nucleofector II apparatus (Lonza) and a MEF1 kit (Lonza – VPD-1004) and T20 program with 5 μg of total plasmid DNA. Cells were first grown to 90% confluency, then 2E6 cells were spun down in 15 ml Falcon tubes at $150 \times g$ for 5 min prior to resuspension in electroporation buffer. After electroporation, cells were plated onto 6-well plates and left to recover for 48 h prior to sorting. Cells were sorted on a BD Influx™ instrument, by the University of Cambridge CIMR sorting facility. eGFP signal on pTracer backbone plasmids was detected by excitation using a 488 nm laser with detection using 530/50 band pass filter. mCherry on pcmCherry backbone plasmids was detected by excitation using a 561 nm laser with detection using 610/20 band pass filter. Positive cells were sorted into DMEM, before being plated onto plates of appropriate area depending on the number of cells sorted.

### Protein extraction from cells or tissues

To extract proteins from cells, cultured HEK293T cells grown in confluent 6-well plates, were first rinsed with ice-cold PBS then lysed with 150 μL RIPA buffer (150 mM NaCl, 1.0% NP-40, 0.5% sodium deoxycholate, 0.1% SDS, 50 mM Tris, pH 8.0) containing 1X cOmplete™ mini EDTA-free Protease Inhibitor Cocktail (Roche - 04693132001) and incubated on ice for 20 min, with occasional agitation. Samples were pipetted up and down, transferred into 1.5 ml Eppendorf tubes then centrifuged ($20,000 \times g$ for 20 min at 4 °C). Supernatant was transferred to a new 1.5 ml Eppendorf tubes and protein concentration determined by BCA quantification (Thermo-Fisher - 23250).

To extract proteins from mouse tissues, frozen mouse samples (~50 mg) were suspended in 500 μL of ice-cold RIPA buffer and homogenized with a gentleMACS™ Dissociator using gentleMACS M tubes (Miltenyi Biotec - 130-093-236) with the recommended setting per tissue type. An initial spin ($500 \times g$ for 5 min at 4 °C) was performed to remove debris and the supernatant was transferred to a new 1.5 ml Eppendorf. Samples were then treated

as above. Resulting samples were stored at −20 °C or used for western blotting immediately.

### Western blotting

Protein lysates (between 10 and 40 µg) were mixed with 10X NuPAGE™ sample reducing agent and 4X NuPAGE™ LDS sample buffer (Invitrogen™ - NP0009 and NP0007, respectively) and incubated for 5 min at 95 °C. Protein samples were loaded and resolved under denaturing conditions by SDS-PAGE using 4–12% gradient precast polyacrylamide NuPAGE® Bis-Tris gels (Invitrogen™). NuPAGE® MES SDS Running Buffer (1X: 50 mM MES, 50 mM Tris Base, 0.1% SDS, 1 mM EDTA, pH 7.3) was used in all applications (Invitrogen™ - NP0002). Gels were run at 200 V for 45 min at RT, with 7 µl Precision Plus Kaleidoscope™ prestained protein Standard (Bio-Rad 1610375). Resulting gels were transferred to PVDF membranes (Invitrogen™ - IB24001) using an iBlot 2 transfer cell (Life Technologies). The resulting PVDF membranes were blocked with 5% milk in PBST (1X PBS with 0.1% Tween 20 (PBS-T)) for 1 h at RT on tube roller and then incubated overnight with specific primary antibodies in 5% milk in PBST at 4 °C (HA, 1:500, Rat, Roche - 11867423001), (FLAG, 1:2000, Mouse, Sigma - F1804), (Vinculin, 1:1000, Mouse, Thermo Fisher - MA5-11690).

Membranes were subsequently washed three times with PBST (10 min at RT) on a tube roller. Then membranes were then incubated with horseradish peroxidase (HRP) - conjugated secondary IgG antibodies in 5% milk in PBST at RT for one hour. (Anti-Rat IgG, 1:1000, Goat, Santa Cruz SC2065), (Anti-mouse IgG, 1:2000, Goat, Promega - W4021).

Two additional washes in PBST were performed (10 min at RT) and a final wash PBS before the membranes were developed by incubating with Amersham ECL™ Western Blotting Detection Reagents (GE Healthcare - RPN2105), per manufacturer's recommendations on an Amersham™ Imager 600.

### Animal models preparation and use

All animal experiments were carried out in accordance with the UK Animals (Scientific Procedures) Act 1986 (PPL70/7538) and EU Directive 2010/63/EU. The C57BL/6j-tRNA[Ala] mice used in this study were housed (from one to four per cage) in a temperature controlled (21 °C) room with a 12 h light-dark cycle and 60% relative humidity. The experimental design included male and female mice, between harboring between 40% and 85% m.5024T. All mice were injected between 8 and 12 weeks of age and were sacrificed 65 days post administration. Mice treated by local intramuscular injection were injected with a 50 µl of vehicle solution (rAAV dilution buffer with 2E11vg HA-eGFP AAV9) into left legs or 50 µl of mtZFN encoding AAV into right legs. The operator was blinded to which injection was administered to the right leg.

Mice treated by systemic administration were injected with 100 µL oh either rAAV dilution buffer in control or AAV solution in experimental conditions (titers ranging from 2E10 to 2.5E12 viral genomes). Operator was blinded to injection conditions.

### DNA extraction and quantification

DNA was extracted from cultured cells or whole tissues using a Qiagen DNEasy Blood & Tissue kit, per the manufacturer's instructions. DNA concentrations were assessed by NanoDrop™ 8000 Spectrophotometer.

### Heteroplasmy analysis by pyrosequencing

Pyrosequencing on m.5024C>T and m.5019A>G mice was performed using the same assay as described in (Nash et al, 2023). Approximately 25 ng of DNA sample was first amplified following the Pyromark® PCR kit (QIAGEN – 978703), per the manufacturer's standard protocol and using the PCR F and PCR R primers indicated in Table 1. Following PCR, 3 µl of Pyromark® magnetic beads (QIAGEN - 974203) were loaded onto Pyromark® Q48 disks (QIAGEN - 974901) and were resuspended with 10 µl of biotinylated PCR product. Disks were run on a Q48 Pyromark sequencer (QIAGEN) using an allele quantification assay, with the following dispensation order: TG/AAGGAC/TTGTAAG and the sequencing primer indicated in Table 1.

### Copy number analysis by qPCR

mtCN was determined by quantitative PCR using PowerUp SYBR Green Master Mix per the manufacturer's protocol (Applied Biosystems - A25742). A 20 µl reaction was used, with 10 ng of DNA template and 500 nM of forward and reverse primer. Samples were analyzed using a QuantStudio™ 3 Real-Time PCR System (Thermo Fisher). The Murine Hexokinase (HK2) was used as a nDNA reference gene and was amplified using HK2 F and HK2 R primers indicated in Table 1. ND1 was used as the mtDNA reference gene and was amplified using ND1 F and ND1 R primers indicated in Table 1. Copy number was then calculated using the $2^{-\Delta\Delta CT}$ Method (Livak and Schmittgen, 2001).

### RNA extraction, northern blotting, and RT-qPCR

Total RNA was extracted from the indicated tissues using the TRIzol™ Reagent (ThermoFisher - 15596026) following the manufacturer's protocol. Frozen mouse tissues (~50 mg) were suspended in 1 ml of TRIzol Reagent and homogenized with a gentleMACS™ Dissociator (Miltenyi Biotec, Bergisch Gladbach, Germany) with the recommended program for each tissue type. A first spin ($500 \times g$ for 5 min at 4 °C) was performed to remove larger debris before supernatant was transferred to a 1.5 ml Eppendorf. A volume of 0.2 ml of chloroform was added per 1 ml of TRIzol, followed by brief vortexing of the tube. Samples were then centrifuged ($15,000 \times g$ for 15 min at 4 °C). The upper aqueous phase was then transferred to a new tube, followed by isopropanol precipitation using 0.5 ml isopropanol per 1 ml of TRIzol used, and incubating at room temperature for 10 min. The tubes were then spin down ($12,000 \times g$ for 10 min at 4 °C) and the supernatant discarded. The pellet was washed twice with ice-cold 70% ethanol and spin down ($7500 \times g$ for 10 min at 4 °C). Supernatant was discarded, the RNA pellet air-dried and finally eluted in nuclease-free water. RNA concentrations were measured by NanoDrop™ 8000 Spectrophotometer (ThermoFisher) at λ = 260/280.

### Detection of mt-tRNA abundance by northern blotting

After quantification, 5 µg of total RNA was resolved on a 10% (w/v) polyacrylamide gel containing 8 M urea. Gels were dry blotted onto a positively charged nylon Hybond-N+ membrane (Cytiva - RPN303B), with the resulting membrane cross-linked by exposure to 254 nm UV light, 120 mJ/cm². For tRNA probes, cross-linked membranes were hybridized with α[32P]-UTP (Hartman – SRP210) radioactively labeled RNA probes T7 transcribed from PCR fragments corresponding to appropriate regions of mouse mtDNA, using MT-TA Forward, MT-TA Reverse primers for mt-tRNA[Ala]

**Table 1. Primer sequences.**

| Primer name | Sequence (5′–3′) |
| --- | --- |
| Nuclear off target analysis by Illumina MiSeq | |
| NGS Forward (NGS_5024_F) | TCGTCGGCAGCGTCAGATGTGTATAAGAGACAGGCATTCAATAGATGTGGG |
| NGS Reverse (NGS_5024_R) | GTCTCGTGGGCTCGGAGATGTGTATAAGAGACAGATGAGTACAATAACCCTACC |
| mtDNA copy number by qPCR | |
| mtDNA HK2 Forward (HK2 F) | GCCAGCCTCTCCTGATTTTAGTGT |
| mtDNA HK2 Reverse (HK2 R) | GGGAACACAAAAGACCTCTTCTGG |
| mtDNA ND1 Forward (ND1 F) | CCGGCTGCGTATTCTACGTT |
| mtDNA ND1 Reverse (ND1 R) | CTAGCAGAAACAAACCGGGC |
| Transcription analysis of immune responses by RT-qPCR | |
| Isg20 Forward | GGCACTGAGACAGGGCTTT |
| Isg20 Reverse | GAGGCCACTCACCCTTTGAG |
| Cxcl10 Forward | CCACGTGTTGAGATCATTGCC |
| Cxcl10 Reverse | TCACTCCAGTTAAGGAGCCC |
| Tnf Forward | GATCGGTCCCCAAAGGGATG |
| Tnf Reverse | TGAGAAGATGATCTGAGTGTGAG |
| Il1b Forward | TGCCACCTTTTGACAGTGATG |
| Il1b Reverse | TGATGTGCTGCTGCGAGATT |
| Ifnb1 Forward | AACTCCACCAGCAGACAGTG |
| Ifnb1 Reverse | GGTACCTTTGCACCCTCCAG |
| Ddx58 Forward | TCTCTTCGTGAAGACCAGAGC |
| Ddx58 Reverse | GGAGCGTCATTCCTGTTGCC |
| mtDNA heteroplasmy analysis by Pyrosequencing | |
| PCR Forward (PCR F) | ATATACTAGTCCGCGAGCCTTC |
| PCR Reverse (PCR R) | [Biotin]–GCAAATTCGAAGGTGTAGAGAAA |
| Sequencing Primer | AAGTTTAACTTCTGATAAGG |
| Probe generation for mt-tRNA abundance measurement by Northern Blot | |
| MT-TA Forward | TAATACGACTCACTATAGGGAGACTAAGGACTGTAAGACTTCATC |
| MT-TA Reverse | GAGGTCTTAGCTTAATTAAAG |
| MT-TW Forward | TAATACGACTCACTATAGGGAGACCAGAAGTTAAACTTGTGTG |
| MT-TW Reverse | AGAAGTTTAGGATATACTAG |

and MT-TW Forward and MT-TW Reverse primers for mt-tRNA$^{Trp}$ (Table 1). Membranes were exposed to a storage phosphor screen and scanned using a Typhoon phosphor imaging system (GE Healthcare). The signals were quantified using Fiji software.

### Transcription analysis of immune responses by RT-qPCR

All RNA samples were first passed through gDNA elimination columns taken from RNeasy kits (QIAGEN - 74134). Subsequently, 2 µg of column purified RNA sample was converted to cDNA using the using the Omniscript Reverse Transcription (RT) Kit (QIAGEN - 205111) using the following reaction components: 2 µg template RNA in a 20 µl reaction, 1X Buffer RT, 0.5 mM of each dNTPs, 10 µM of oligo dT primer (Invitrogen), 0.2 U/µl of reverse transcriptase, 10 U/µl of RNAsin® RNase inhibitor (Promega - N2515), nuclease-free H$_2$O to 20 µL. The final mixtures were incubated for 1 h at 37 °C. Resulting cDNA was diluted 10-fold in nuclease-free water prior to qPCR using PowerUp SYBR

Green Master Mix per the manufacturer's protocol (Applied Biosystems - A25742). Primer pairs for each transcript are provided in Table 1.

### Next-generation sequencing and off-target analysis

The region flanking the mtZFN target site was aligned to the *M. musculus* reference genome (GRCm39) yielding two regions of near perfect homology to the target window. A 149-bp amplicon was amplified from the DNA samples extracted from skeletal muscle using Phusion® DNA polymerase (NEB – M0536S) and NGS Forward and NGS Reverse primers with adapter sequences.

Obtained PCR amplicons were subjected to NexteraXT sample processing, and the resulting libraries were assessed by 2×150-cycle paired-end sequencing using a NovaSeq instrument (Illumina). Quality trimming and 3′-end adapter clipping of sequenced reads were performed simultaneously with Trim Galore! (--paired) and

**The paper explained**

**Problem**

Mutations in mitochondrial DNA (mtDNA) are linked to various untreatable diseases. Current gene therapy approaches using mito-chondrially targeted nucleases, while promising, face challenges including high-dose requirements and safety concerns. Developing efficient, tissue-specific, and clinically translatable therapies is critical to address these limitations.

**Results**

This study demonstrates a novel approach for mitochondrial gene therapy using a tandem zinc finger nuclease (mtZFN) architecture delivered via adeno-associated viral (AAV) vectors. The tandem mtZFNs allow efficient packaging into a single AAV, reducing the mtDNA mutation load in cardiac and skeletal muscle tissues in a mouse model. This approach showed significant molecular phenotypic rescue of the pathogenic mutation effects. Key results include:

- Improved efficiency of heteroplasmy modification using the tandem architecture compared to traditional dual-viral delivery.
- Reduction in effective dose requirements by up to tenfold.
- Limited off-target effects on nuclear DNA and minimal immune responses.

**Impact**

The findings highlight the therapeutic potential of tandem mtZFNs as a safer and more effective gene therapy platform for mitochondrial dis-eases. By overcoming previous technical barriers and improving safety, this method represents a significant step towards clinical application, paving the way for targeted and systemic treatments of mtDNA dis-orders.

aligned to GRCm39 using bowtie2. Reads that contained the entire region of chromosome 5 (60200118–60200266) or chromosome 2 (22480080–22479932) were selected for counting with SAMtools (flagstat) and insertion/deletion count based on CIGAR string (I/D). All individual samples yielded >20,000 reads per nucleotide.

## Statistics

In vivo experiments had operators blinded as to the treatment conditions. All experiments included a minimum of 4 biological replicates, with exact values written in the figure legends. The statistical tests used were specific to the setup of each experiment and are indicated in corresponding figure legends with all significant $p$-values indicated on the figures themselves.

## Data availability

The datasets produced in this study are available in the following databases: The NGS data: Gene Expression Omnibus (GEO) GSE291762.

The source data of this paper are collected in the following database record: biostudies:S-SCDT-10_1038-S44321-025-00231-5.

## Peer review information

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

## Acknowledgements

This research has been unded by UKRI Medical Research Council grants MC_UU_00015/4 and MC_UU_00028/3 (PAN, KMT, CAP, LVH, PS-P, PAG, and MM), UKRI MRC award MC_PC_21046 to the National Mouse Genetics Network Mitochondria Cluster (MitoCluster) (PAN, PS-P, KMT, CAP, LVH, and MM), The Lily Foundation (PAN), The Champ Foundation (PS-P), CRUK SI Core Funding A31287 (PAG), A_BICR_1920_Gammage (PAG). We acknowledge the significant contribution to model development made by Dr. James B. Stewart (Newcastle University, UK) and Prof. Nils-Goran Larsson (Karolinska Institute, Stockholm, Sweden), which was essential to this work. We are grateful to the personnel at Phenomics Animal Care Facility and The Anne McLaren Building for their technical support in managing the mouse colonies. We are grateful to Martin Rice, Phenomics Animal Care Facility, Cambridge, for technical assistance with viral administration. The FACS experiments were performed by the CIMR Flow facility to whom we are very grateful.

## Author contributions

**Pavel A Nash**: Conceptualization; Data curation; Formal analysis; Validation; Investigation; Visualization; Methodology; Writing—original draft; Project administration; Writing—review and editing. **Keira M Turner**: Investigation; Writing—review and editing. **Christopher A Powell**: Formal analysis; Investigation; Methodology; Writing—review and editing. **Lindsey Van Haute**: Data curation; Software; Formal analysis; Investigation; Methodology; Writing—review and editing. **Pedro Silva-Pinheiro**: Investigation; Writing—review and editing. **Felix Bubeck**: Investigation; Writing—review and editing. **Ellen Wiedtke**: Investigation; Writing—review and editing. **Eloïse Marques**: Investigation; Writing—review and editing. **Dylan G Ryan**: Resources; Formal analysis; Supervision; Funding acquisition; Investigation; Methodology; Project administration; Writing—review and editing. **Dirk Grimm**: Resources; Supervision; Methodology; Project administration; Writing—review and editing. **Payam A Gammage**: Conceptualization; Data curation; Formal analysis; Funding acquisition; Investigation; Methodology; Writing—original draft; Project administration; Writing—review and editing. **Michal Minczuk**: Conceptualization; Resources; Data curation; Supervision; Funding acquisition; Validation; Visualization; Writing—original draft; Project administration; Writing—review and editing.

Source data underlying figure panels in this paper may have individual authorship assigned. Where available, figure panel/source data authorship is listed in the following database record: biostudies:S-SCDT-10_1038-S44321-025-00231-5.

## Disclosure and competing interests statement

MM is a co-founder, shareholder, and member of the Scientific Advisory Board of Pretzel Therapeutics, Inc. PAG is a shareholder and provided consultancy services for Pretzel Therapeutics, Inc. PS-P and PAN provided consultancy services for Pretzel Therapeutics, Inc. LVH is director of NextGenSeek Ltd. The remaining authors declare no competing interests. PAG and MM are authors of a patent application WO2020188228A1 pertaining to the optimization and delivery of mitochondrial proteins in a single expression vector.

