## [Peer Review File · EMBO Molecular Medicine]

Clinically translatable mitochondrial gene therapy in muscle using tandem mtZFN architecture

Michal Minczuk, Pavel Nash, Keira Turner, Christopher Powell, Lindsey Van Haute, Pedro Silva-Pinheiro, Felix Bubeck, Ellen Wiedtke, Eloïse Marques, Dylan Ryan, Dirk Grimm, and Payam Gammage

Corresponding author: Michal Minczuk (mam@mrc-mbu.cam.ac.uk)

Review Timeline:

Submission Date:	8th Oct 24
Editorial Decision:	6th Nov 24
Revision Received:	13th Mar 25
Accepted:	14th Mar 25

Editor: Zeljko Durdevic

Transaction Report:

6th Nov 2024

Dear Prof. Minczuk,

Thank you for the submission of your manuscript to EMBO Molecular Medicine. We have now received feedback from the three reviewers who agreed to evaluate your manuscript. As you will see from their reports pasted below all three referees support publication of your manuscript raising important but minor concerns. Therefore, I am pleased to inform you that we will be able to accept your manuscript pending the following final amendments:

1) Please address all referee concerns. No additional experiments are required. Particular attention should be given to streamlining the discussion and rationalizing the number of figures (potentially some figures could be merged, e.g. Fig1-3 and Fig5-6). Please consider reformatting your manuscript to a Report type of article (3-4 figures, ~22000 characters) as suggested by the referee #2, for more information please check our "Author Guidelines".

<https://www.embopress.org/page/journal/17574684/authorguide#reportsarticleguide>

2) Reorder manuscript sections: Abstract / Keywords / Introduction / Results / Discussion / Methods / Acknowledgements / Disclosure and competing interests statement / The Paper Explained / References / Figure legends

3) Author checklist: Please submit a complete checklist. <https://www.embopress.org/pb-assets/embosite/EMBO%20Press%20Author%20Checklist-1642513524327.xlsx>

4) In the main manuscript file, please do the following:

- Please address all comments suggested by our data editors listed below:

o Figure legends:

1. Please note that the figure 6 does not contain a box plot, kindly rectify the box plot related information in the figure legend appropriately.

2. Please note that the legends for figures 6A-F are not bifurcated. This needs to be rectified.

3. Please note that the exact p values are not provided in the legends of figures 2B, 3B, C; 4B, D; 5B; 6A, E, F; 7B

4. Please indicate the statistical test used for data analysis in the legends of figures 3B, 4B, 5B; EV5 A, B

5. Please note that in figures 2B, 3B, C; 5B, 6A, E, F; 7B there is a mismatch between the "****/**/**/" in the figure legend and the "****/**/**/" in the figure, that should be corrected.

6. Please note that the box plots need to be defined in terms of minima, maxima, centre, bounds of box and whiskers, and percentile in the legends of figures EV5 A, B.

7. Please note that information related to n is missing in the legends of figures 3B, C; 4B, C; 5B, C; 6A-F; 7B, C, D; EV2 B; EV5 A, B; EV6.

8. Please note that the error bars are not defined in the legends of figures 3B, 4B, 5B, 6A-F; 7B, C; EV 2B; EV6.

- Add callouts for Fig 3C, Fig 5B, Fig 6B, C, D.

- In Methods, add statistical paragraph that should reflect all information that you have filled in the Authors Checklist, especially regarding randomization, blinding, replication etc.

- Indicate in legends exact n and exact p values, not a range, along with the statistical test used. To keep the figures "clear" some authors found providing an Appendix table Sx with all exact p-values preferable. You are welcome to do this if you want to.

- Please include structured Methods section that includes a Reagents and Tools Table (should be uploaded as a separate file) followed by a Methods and Protocols section. More information on how to adhere to this format as well as downloadable templates (.docx) for the Reagents and Tools Table can be found in our author guidelines:

<https://www.embopress.org/page/journal/17574684/authorguide#structuredmethods>

An example of a paper with Structured Methods can be found here:

<https://www.embopress.org/doi/full/10.1038/s44320-024-00037-6#sec-4>

- Author contributions: Please remove it from the manuscript and specify author contributions in our submission system. CRediT has replaced the traditional author contributions section because it offers a systematic machine-readable author contributions format that allows for more effective research assessment. You are encouraged to use the free text boxes beneath each contributing author's name to add specific details on the author's contribution. More information is available in our guide to authors:

<https://www.embopress.org/page/journal/17574684/authorguide#authorshipguidelines>

- Please remove the sentence "All data are available in the main text or the EV materials" data availability statement. Please be aware that raw data from large-scale datasets (Next generation sequencing) should be deposited in one of the relevant databases and made freely available prior the publication of the manuscript. Use the following format to report the accession number of your data:

[data type]: [full name of the resource] [accession number/identifier] ([doi or URL or identifiers.org/DATABASE:ACCESSION])

Please check "Author Guidelines" for more information.

<https://www.embopress.org/page/journal/17574684/authorguide#availabilityofpublishedmaterial>

- Correct the reference citation in the reference list. Where there are more than 10 authors on a paper, 10 will be listed, followed by "et al.". Please check "Author Guidelines" for more information.

<https://www.embopress.org/page/journal/17574684/authorguide#referencesformat>

5) Funding: Please merge it with "Acknowledgements" and make sure that information about all sources of funding are complete in both our submission system and in the manuscript. Currently, CRUK SI Core Funding A31287 and A_BICR_1920_Gammage is missing in our submission system.

6) Appendix: Please rename "Expanded View Content" to "Appendix" and correct the nomenclature to "Appendix Figure S1" etc. in the file and in the manuscript text. Remove their legends from the main manuscript file and add them to the appendix file under the corresponding figure. Please rename the "Alignment" shown on pages 8 - 11 to "Appendix Supplementary Data" and update its callout in the main text. On the title page add table of content with page numbers.

7) Dataset: Uploaded dataset seems to contain numerical source data underlying figures. This should be submitted as source data, which will be requested by our source data editor Hannah Sonntag. You will receive a separate e-mail with instructions and source data checklist.

8) The Paper Explained: Please provide "The Paper Explained" and add it to the main manuscript text. Please check "Author Guidelines" for more information. <https://www.embopress.org/page/journal/17574684/authorguide#researcharticleguide>

9) Synopsis: Every published paper now includes a 'Synopsis' to further enhance discoverability. Synopses are displayed on the journal webpage and are freely accessible to all readers. They include separate synopsis image and synopsis text.

- Synopsis text: Please remove the bullet points from the main manuscript text and provide a short standfirst (maximum of 300 characters, including space) as well as 2-5 one sentence bullet points that summarise the paper as a .doc file. Please write the bullet points to summarise the key NEW findings. They should be designed to be complementary to the abstract - i.e. not repeat the same text. We encourage inclusion of key acronyms and quantitative information (maximum of 30 words / bullet point). Please use the passive voice.

- Synopsis image: Please provide a visual abstract as a high-resolution jpeg file 550 pixels wide x 200-600 pixels high to illustrate your article.

10) As part of the EMBO Publications transparent editorial process initiative (see our Editorial at <http://embomolmed.embopress.org/content/2/9/329>), EMBO Molecular Medicine will publish online a Review Process File (RPF) to accompany accepted manuscripts. This file will be published in conjunction with your paper and will include the anonymous referee reports, your point-by-point response and all pertinent correspondence relating to the manuscript. Let us know whether you agree with the publication of the RPF and as here, if you want to remove or not any figures from it prior to publication. Please note that the Authors checklist will be published at the end of the RPF.

11) Please provide a point-by-point letter INCLUDING my comments as well as the reviewer's reports and your detailed responses (as Word file).

I look forward to reading a new revised version of your manuscript as soon as possible.

*** Instructions to submit your revised manuscript ***

1) a .docx formatted version of the manuscript text (including Figure legends and tables)

2) Separate figure files*

3) supplemental information as Expanded View and/or Appendix. Please carefully check the authors guidelines for formatting Expanded view and Appendix figures and tables at <https://www.embopress.org/page/journal/17574684/authorguide#expandedview>

4) a letter INCLUDING the reviewer's reports and your detailed responses to their comments (as Word file).

5) The paper explained: EMBO Molecular Medicine articles are accompanied by a summary of the articles to emphasize the major findings in the paper and their medical implications for the non-specialist reader. Please provide a draft summary of your article highlighting

6) Author contributions: the contribution of every author must be detailed in a separate section.

7) EMBO Molecular Medicine now requires a complete author checklist (<https://www.embopress.org/page/journal/17574684/authorguide>) to be submitted with all revised manuscripts. Please use the checklist as guideline for the sort of information we need WITHIN the manuscript. The checklist should only be filled with page numbers where the information can be found. This is particularly important for animal reporting, antibody dilutions (missing) and exact values and n that should be indicated instead of a range.

8) Every published paper now includes a 'Synopsis' to further enhance discoverability. Synopses are displayed on the journal webpage and are freely accessible to all readers. They include a short stand first (maximum of 300 characters, including space) as well as 2-5 one sentence bullet points that summarise the paper. Please write the bullet points to summarise the key NEW findings. They should be designed to be complementary to the abstract - i.e. not repeat the same text. We encourage inclusion of key acronyms and quantitative information (maximum of 30 words / bullet point). Please use the passive voice. Please attach these in a separate file or send them by email, we will incorporate them accordingly.

You are also welcome to suggest a striking image or visual abstract to illustrate your article. If you do please provide a jpeg file 550 px-wide x 300-600px high.

9) A Conflict of Interest statement should be provided in the main text

10) Please note that we now mandate that all corresponding authors list an ORCID digital identifier. This takes <90 seconds to complete. We encourage all authors to supply an ORCID identifier, which will be linked to their name for unambiguous name identification.

Currently, our records indicate that the ORCID for your account is 0000-0001-8242-1420.

Link Not Available

11) Include a Reagents and Tools Table as part of the Methods section, which can be downloaded from our author guidelines (<https://www.embopress.org/page/journal/17574684/authorguide#structuredmethods>)

Photos 400-800 DPI

*Additional important information regarding figures and illustrations can be found at

***** Reviewer's comments *****

Referee #1 (Comments on Novelty/Model System for Author):

This is a very well conducted study showing how a novel design of zinc finger nucleases can be used to shift heteroplasmy level in post-mitotic tissue. Mitochondrial diseases stemming from mutations in mtDNA are quite common and hard to treat, the here-developed technique is therefore a milestone in finding cures to these diseases that affect post-mitotic tissues.

Referee #1 (Remarks for Author):

This is a very good paper on the technology development to correct heteroplasmy levels, where WT mtDNA and a pathogenic mutated form coexist. By targeting the pathogenic variant with assembling a zinc finger nuclease at the site of mutation, heteroplasmy levels could be corrected. Moreover, two sets of approaches for the delivery were examined and both worked, with minimal off-target effects and immune activation. The manuscript is well written, but could benefit from a discussion about the use of the 2A peptide and why the authors prefer this over a strategy employing an internal MPP site, which should do the same job, but further reduce the size of the necessary coding region.

Referee #2 (Remarks for Author):

This is a neat report on the optimisation of ZFN-mediated modulation of mtDNA heteroplasmy. In particular, the authors report data to show that constructs can be made that express an entire ZFN duplex from a single AAV genome. The long term goal of this approach is clear - to be able to use AAV-mediated heteroplasmy depletion in patients with mitochondrial disease. The authors use a variety of viral serotypes on a heteroplasmic mouse model of mito disease to show cardiac or muscle targeting, with the resultant heteroplasmy levels being reduced for the mt-tRNAAla mouse. MtDNA copy number was assessed as well as the relative steady state levels of mt-tRNAAla. The approach was clearly successful and the manuscript includes explanations of how the method has been optimised. Overall, I found this very convincing work. I have a couple of points the authors may wish to address. First, I found the discussion section rather excessive, repeating much of the results section. Would be good to see that more focussed. This did make me wonder whether the report would be better as a short report, as the data is simple and clear? Second, I found the comment at the end of the discussion about the relative titre levels for use in humans still being prohibitive to be very interesting. Perhaps the authors could expand this a little, as it is very relevant to therapy. Finally, what might be the advantages/disadvantages of using the mtZFN approach when compared to mitoARCUS ?

Referee #3 (Remarks for Author):

This is an interesting paper which looks at the practicality of treating heteroplasmic mtDNA diseases with zinc finger nucleases (ZFNs). This is a follow on paper from the Gammage et al manuscript which showed that ZFNs could decrease the pathogenic variant level in a heteroplasmic mouse model. They have performed some experiments to determine whether this could be translated by developing tandem ZFNs

1. I think it is a very good idea to develop this technology and they have to my mind convincingly shown it will work, at least in the animal model.
2. They are more than well aware of some of the risks involved - for example lowering mtDNA copy number and the risks of an immune reaction.
3. They are also aware that whilst they have altered the genotype because the mouse has little in the way of clinical or biochemical phenotype, they have not proved it will improve clinical disease. However, since we know in these diseases that heteroplasmy level one of the most important factors then I can see no reason why it would not be effective.
4. The difficulty with heteroplasmic mtDNA disorders is that they are almost always systemic with multiple tissues involved, apart perhaps from some patients with single deletion disease and some very rare patients with pathogenic variants and a pure muscle phenotype for example. Thus, any treatment that they develop will almost certainly have to reach multiple organs. The fact that they can use tandem ZFNs is impressive but still does not get around this difficult problem. However, this is an important step forward

Referee #1 (Comments on Novelty/Model System for Author):

This is a very well conducted study showing how a novel design of zinc finger nucleases can be used to shift heteroplasmy level in post-mitotic tissue. Mitochondrial diseases stemming from mutations in mtDNA are quite common and hard to treat, the here-developed technique is therefore a milestone in finding cures to these diseases that affect post-mitotic tissues.

We are thankful for this highly positive evaluation and for recognising the significance of our work.

Referee #1 (Remarks for Author):

This is a very good paper on the technology development to correct heteroplasmy levels, where WT mtDNA and a pathogenic mutated form coexist. By targeting the pathogenic variant with assembling a zinc finger nuclease at the site of mutation, heteroplasmy levels could be corrected. Moreover, two sets of approaches for the delivery were examined and both worked, with minimal off-target effects and immune activation. The manuscript is well written, but could benefit from a discussion about the use of the 2A peptide and why the authors prefer this over a strategy employing an internal MPP site, which should do the same job, but further reduce the size of the necessary coding region.

Once again, we would like to express our gratitude to the reviewer for their positive comments. The use of 2A peptides is common in gene therapy applications involving AAV vectors (doi.org/10.1038/s44319-024-00244-0), which is the main reason we chose them for our application. However, using an internal MPP site as an alternative is an intriguing suggestion that potentially

deserves further investigation. To our knowledge, there is no documented use of an internal MPP site in recombinant proteins for mitochondrial co-delivery. Literature suggests that mouse *Sdha* is cleaved more than 100 amino acids from the N-terminus (doi: 10.1074/mcp.M116.063818). It would be essential to test whether such a cleavage can occur more than 500 amino acids from the N-terminus. We've added a sentence to the discussion to address this potential avenue for further research.

Referee #2 (Remarks for Author):

This is a neat report on the optimisation of ZFN-mediated modulation of mtDNA heteroplasmy. In particular, the authors report data to show that constructs can be made that express an entire ZFN duplex from a single AAV genome. The long-term goal of this approach is clear - to be able to use AAV-mediated heteroplasmy depletion in patients with mitochondrial disease. The authors use a variety of viral serotypes on a heteroplasmic mouse model of mito disease to show cardiac or muscle targeting, with the resultant heteroplasmy levels being reduced for the mt-tRNA^{Ala} mouse. MtDNA copy number was assessed as well as the relative steady state levels of mt-tRNA^{Ala}. The approach was clearly successful and the manuscript includes explanations of how the method has been optimised. Overall, I found this very convincing work.

We appreciate this positive evaluation.

I have a couple of points the authors may wish to address. First, I found the discussion section rather excessive, repeating much of the results section. Would be good to see that more focussed. This did make me wonder whether the report would be better as a short report, as the data is simple and clear?

We have removed redundancy in the discussion where possible to make it lighter. We appreciate the comment about the clarity of our data and that the paper could be re-written as a short report. We feel that much of the clarity stems from the data being shown in a highly visual and stepwise manner, with distinct separations between experimental design, in vitro and in vivo data.

Second, I found the comment at the end of the discussion about the relative titre levels for use in humans still being prohibitive to be very interesting. Perhaps the authors could expand this a little, as it is very relevant to therapy.

The comment about the viral titres being too high refers to our previously published study with separate viral vectors (Gammage et al. 2018). This current study reduces the effective viral titre to levels comparable to human clinical trials as shown in Appendix figure 7. We have reworded the relevant paragraph to clarify this issue in the revised version of the manuscript.

Finally, what might be the advantages/disadvantages of using the mtZFN approach when compared to mitoARCUS ?

As zinc fingers are ubiquitous mammalian transcription factors, they have inherently lower immunogenicity, and regulatory hurdles for drug development are likely to be reduced. Furthermore, there is publicly available knowledge on zinc finger array design, enabling researchers to design their own mtZFNs which is not the case for I-Cre meganucleases, such as mitoARCUS. In terms of performance in heteroplasmy shifting mitoARCUS shows higher shifts in heteroplasmy in the same model as this study (Zekonyte, 2021), though we believe that by improving the specificity of our ZF arrays, a similar effect can be achieved.

Referee #3 (Remarks for Author):

This is an interesting paper which looks at the practicality of treating heteroplasmic mtDNA diseases with zinc finger nucleases (ZFNs). This is a follow on paper from the Gammage et al manuscript which showed that ZFNs could decrease the pathogenic variant level in a heteroplasmic mouse model. They have performed some experiments to determine whether this could be translated by developing tandem ZFNs

1. I think it is a very good idea to develop this technology and they have to my mind convincingly shown it will work, at least in the animal model.
2. They are more than well aware of some of the risks involved - for example lowering mtDNA copy

number and the risks of an immune reaction.

3. They are also aware that whilst they have altered the genotype because the mouse has little in the way of clinical or biochemical phenotype, they have not proved it will improve clinical disease.

However, since we know in these diseases that heteroplasmy level one of the most important factors then I can see no reason why it would not be effective.

4. The difficulty with heteroplasmic mtDNA disorders is that they are almost always systemic with multiple tissues involved, apart perhaps from some patients with single deletion disease and some very rare patients with pathogenic variants and a pure muscle phenotype for example. Thus, any treatment that they develop will almost certainly have to reach multiple organs. The fact that they can use tandem ZFNs is impressive but still does not get around this difficult problem. However, this is an important step forward

We thank the reviewer for the positive feedback on our work. The issues the reviewer raised are, of course, correct. The further implementation of this technology for human treatment will additionally require further development in tissue-specific delivery methods such as further AAV capsid refinement or developments in nanoparticle technology. The development of such technologies is being pursued in specialised laboratories. Indeed, the AAVMYO capsid used in the later part of our work is a capsid obtained in collaboration with such a specialized laboratory. A sentence reflecting this fact has been added to the discussion. We are currently also writing a review article that raises these exact points.

14th Mar 2025

Dear Prof. Minczuk,

We are pleased to inform you that your manuscript is accepted for publication and is now being sent to our publisher to be included in the next available issue of EMBO Molecular Medicine.

Zeljko Durdevic
Senior Editor
EMBO Molecular Medicine
